# Variations in Antibiotic Use and Sepsis Management in Neonatal Intensive Care Units: A European Survey

**DOI:** 10.3390/antibiotics10091046

**Published:** 2021-08-27

**Authors:** Felipe Garrido, Karel Allegaert, Cristina Arribas, Eduardo Villamor, Genny Raffaeli, Miren Paniagua, Giacomo Cavallaro

**Affiliations:** 1Department of Pediatrics, Clínica Universidad de Navarra, 28027 Madrid, Spain; carribass@unav.es (C.A.); mpaniagua.1@alumni.unav.es (M.P.); 2Department of Development and Regeneration, KU Leuven, 3000 Leuven, Belgium; allegaertkarel@hotmail.com; 3Department of Pharmaceutical and Pharmacological Sciences, KU Leuven, 3000 Leuven, Belgium; 4Department of Hospital Pharmacy, Erasmus MC University Medical Center, 3015 GD Rotterdam, The Netherlands; 5Department of Pediatrics, Maastricht University Medical Center (MUMC+), School for Oncology and Developmental Biology (GROW), 6229 Maastricht, The Netherlands; e.villamor@mumc.nl; 6Fondazione IRCCS Ca’ Granda Ospedale Maggiore Policlinico, NICU, 20122 Milan, Italy; genny.raffaeli@unimi.it (G.R.); giacomo.cavallaro@policlinico.mi.it (G.C.); 7Department of Clinical Sciences and Community Health, Università degli Studi di Milano, 20122 Milan, Italy

**Keywords:** antibiotics, neonatal intensive care unit, early-onset sepsis, late-onset sepsis, meningitis

## Abstract

Management of neonatal sepsis and the use of antimicrobials have an important impact on morbidity and mortality. However, there is no recent background on which antibiotic regimens are used in different European neonatal intensive care units (NICUs). Our study aimed to describe the use of antibiotics and other aspects of early- and late-onset sepsis (EOS and LOS, respectively) management by European NICUs. We conducted an online survey among NICUs throughout Europe to collect information about antibiotic stewardship, antibiotic regimens, and general aspects of managing neonatal infections. NICUs from up to 38 European countries responded, with 271 valid responses. Most units had written clinical guidelines for EOS (92.2%) and LOS (81.1%) management. For EOS, ampicillin, penicillin, gentamicin, and amikacin were the most commonly used antibiotics. Analysis of the combinations of EOS regimens showed that the most frequently used was ampicillin plus gentamicin (54.6%). For LOS, the most frequently used antibiotics were vancomycin (52.4%), gentamicin (33.9%), cefotaxime (28%), and meropenem (15.5%). Other aspects of the general management of sepsis have also been analyzed. The management of neonatal sepsis in European NICUs is diverse. There was high self-reported adherence to the local clinical guidelines. There was homogeneity in the combination of antibiotics in EOS but less in LOS.

## 1. Introduction

The neonatal period is a very vulnerable time in life in which sepsis or meningitis is a frequent primary cause of death [1,2,3]. The Global Burden of Disease study estimated that 2202 neonates for every 100,000 live births develop neonatal sepsis, equating to three million cases of neonatal sepsis annually, with a mortality rate of 11–19% [4,5,6]. Group B streptococcus (GBS) is the most common pathogen of neonatal sepsis in high-income countries such as Europe, where the incidence of early-onset GBS disease is 0.45 per 1000 live births [7]. In addition, the incidence of Gram-negative infections continues to increase, along with a high level of antibiotic resistance [8,9,10,11].

In some neonatal intensive care units (NICUs), more than 70% of neonates have been prescribed antibiotics. However, the early use of antibiotics can have undesirable consequences in the clinical course of these babies, decisively determining their prognosis in terms of risk of antibiotic resistance, altered microbiome, interference with mother–child interaction, and high health care costs [10,12,13]. For example, each day of antibiotic therapy has been associated with a 1.24 times increased risk of sepsis, necrotizing enterocolitis (NEC), or death [14].

Recently, tools such as the Kaiser Permanente early-onset sepsis (EOS) calculator have been developed to predict the risk of infection and reduce unnecessary empirical tests and treatments [15,16,17,18]. However, the sensitivity of such calculators is somewhat limited and focuses only on EOS, and is unable to predict the risk of late-onset sepsis (LOS) [19]. Thus, the development of clinical and therapeutic guidelines specifically for local microorganisms can optimize the management of neonatal sepsis and antibiotic use.

Antimicrobial stewardship measures and improves the prescription of antibiotics by clinicians. Enhancing antibiotic prescription and use is crucial to adequately manage infections, shield patients from injuries induced by unnecessary antibiotic use, and combat antibiotic resistance [20,21,22]. However, adherence to antibiotic stewardship may vary significantly between institutions. Ho et al. stated that significant gaps exist between the Centers for Disease Control and Prevention’s (CDC) recommendations to improve antibiotic use and institutions’ antibiotic practices during the neonatal period [23].

Data across Europe on the current practice of antimicrobial use in neonatal sepsis and the general management of sepsis are also limited [1,24,25]. However, previous studies have described some variability in the dosage of commonly used antibiotics in the neonatal period and the existence of clinical guidelines [26,27]. In one European survey, Spyridis et al. reported that 71% of the hospitals had comprehensive antibiotic prescription guidelines for treating sepsis, and recorded 20 different antibiotic regimens for EOS and LOS [28]. 

The aim of our study was to describe the use of antibiotics for the treatment of neonatal infections by European NICUs and characterize the general management of sepsis and perinatal infections across Europe.

## 2. Results

The lack of a complete list of neonatal units and NICUs in Europe did not allow us to determine an ample sample size [29,30,31]. Furthermore, due to the distribution method of the survey, it was not possible to obtain complete information on the number of NICUs contacted. Some neonatologists and presidents of scientific societies have even distributed our surveys in their country without communicating the number of NICUs contacted. Therefore, the response rate cannot be determined.

We obtained a total number of 280 responses to the questionnaire. However, after identifying the respondent hospitals, we detected three responses from non-European countries and duplicated responses from several hospitals, resulting in 271 valid responses (Figure 1).

Neonatal units from up to 38 European countries responded to the survey. Figure 2 shows the countries from which we obtained at least six responses. The rest of the countries from which we obtained five or fewer responses are listed in a table in the Appendix A. Most responding units were level 3 (208, 78.2%). In addition, the majority of the responding NICUs (83.4%) provided care for infants from less than 28 weeks of gestation until discharge, and 48.3% had more than 50 admissions of babies with less than 1500 g in 2020. The demographic details of the responding NICUs are shown in Table 1.

The majority of NICUs had written clinical guidelines for managing EOS (92.2%) and LOS (81.1%). Units that stated that they had clinical guidelines for EOS management reported that their level of adherence was high or very high (93.1%). Reported adherence to LOS guidelines was also high or very high (85.5%). Only 162 NICUs (57.7%) had a clinical guide for suspending antibiotic therapy for neonatal sepsis. The reported adherence to this guideline for treatment suspension for neonatal sepsis was high or very high in 74.6% of NICUs.

We analyzed some aspects of the management of risk factors for sepsis in maternity wards. Most of the maternity wards linked to NICUs stated that they had written a clinical guide to identify and treat mothers at risk of perinatal infection (80.8%). Adherence to these clinical guidelines was considered high or very high in 77.3% of the centers. Regarding GBS detection in mothers at the end of pregnancy, 60.5% of the centers performed universal screening, 21% screened selected mothers, and 18.4% did not test for GBS. The distribution of answers by country can be found in the Appendix A. Concerning the use of Kaiser Permanente-type perinatal infectious risk prediction calculators, 63.1% of those centers did not use them, 22.1% reported regular use, and 14.7% reported occasional use, depending on the neonatologist. 

The NICUs were asked about the antibiotic combinations used for EOS, LOS, and NEC. The results of these questions are summarized in Table 2. For EOS, ampicillin, penicillin, gentamicin, and amikacin were the most commonly used antibiotics. Analysis of the antibiotic combinations for EOS showed that the most frequent was ampicillin plus gentamicin (54.6%), followed by penicillin plus gentamicin (17.3%), and ampicillin plus amikacin (10.3%). For LOS, the most widely used antibiotics were vancomycin (52.4%), gentamicin (33.9%), cefotaxime (28%), and meropenem (15.5%). The combinations used for LOS were less homogeneous, with amikacin plus vancomycin (10.7%), cefotaxime plus vancomycin (8.1%), and meropenem plus vancomycin (7.7%) being the most frequent.

The most commonly used antibiotics in the empirical treatment of NEC were vancomycin (39.5%). gentamicin (35.4%), meropenem (23.6%), and amikacin (22.1%). However, the combinations were not homogeneous, with ampicillin plus gentamicin plus metronidazole (10%) and meropenem plus vancomycin (9.2%) being the most frequent. The rest of the antibiotic combinations can be found in the Appendix A.

Approximately one-third (29.9%) of the NICUs did not perform therapeutic drug monitoring (TDM). In NICUs that performed TDM, the most commonly monitored antibiotics were vancomycin (142; 52.4%), gentamicin (116; 42.8%), and amikacin (60; 22.1%). The reasons for not performing TDM were not investigated in this study.

Regarding the laboratory and biological markers commonly used to diagnose neonatal infections, the most frequent combination was blood culture plus C-reactive protein (CRP) (71 NICUs; 26.8%), followed by a combination of blood culture, CRP, and procalcitonin (PCT) (66 NICUs; 24.4%). On the other hand, the most common laboratory testing combination used to decide when to discontinue antibiotics was blood culture plus CRP (112 NICUs; 41.3%). A summary of the laboratory markers used is shown in Table 3. In addition, all combinations of laboratory markers for the initiation and suspension of antibiotics can be found in the Appendix A.

When we asked about performing a lumbar puncture (LP) and cerebrospinal fluid (CSF) examination in a clinically septic neonate (Table 4), most neonatologists (191 NICUs; 70.5%) answered that the decision to perform these procedures is based on clinical history, risk factors, and the clinical characteristics of the neonate. Only 15.9% of neonatologists always performed an LP if the newborn was clinically stable.

Of the respondents, 70.8% stated that they did not use oral antibiotics. Among those who used oral antibiotics, 61% chose the oral route in cases of limited vascular access, 36% used them as prophylactic treatment in cases with risk of infection, and 61% used them in cases of mild infections.

More than half of the units reported using antifungal prophylaxis in very low-weight preterm newborns (159 NICUs; 58.6%).

## 3. Discussion

Judicious use of antibiotics is crucial for reducing antimicrobial resistance [32]. In this survey, we collected information on (1) the most commonly used antibiotic combinations in the empiric treatment of early and late-onset neonatal sepsis, and (2) the existence of clinical guidelines and adherence to said guidelines in European NICUs. We also addressed some other aspects of perinatal infection management.

Our results showed that the combination of ampicillin or penicillin plus an aminoglycoside was the most frequently used for EOS. However, we detected less homogeneity in the empirical management of LOS or NEC. The World Health Organization (WHO) currently recommends using ampicillin (or penicillin) and gentamicin for the empirical treatment of EOS or LOS [33]. On the other hand, neonates with signs of staphylococcal infection are recommended to receive cloxacillin rather than ampicillin [34]. Liem et al. reported on the use of neonatal antibiotic therapy in 10 NICUs in The Netherlands [35]. They found that the combination of amoxicillin, amoxicillin/clavulanic, or penicillin plus an aminoglycoside (gentamicin or amikacin) was the empirical therapy used for EOS. In addition, they observed high variability in the combination of antibiotics used for LOS. The most commonly used antibiotics were flucloxacillin, gentamicin, ceftazidime, and vancomycin. In contrast, we found increased heterogeneity, with the most combined antibiotics comprising vancomycin, gentamicin, amikacin, cefotaxime, meropenem, and piperacillin-tazobactam.

In our study, the number of NICUs with existing antibiotic guidelines for EOS and LOS was high. The level of reported adherence to these guidelines was high or very high as well. In addition, many NICUs reported having clinical guidelines for initiating empirical antimicrobial treatment or decision making in newborns at infectious risk in maternity wards. In The Netherlands, Liam et al. reported that all NICUs had empirical treatment guidelines for neonatal sepsis. However, only seven had antibiotic guidelines for meningitis and six for NEC [35]. Undoubtedly, this practice should lead to the rational use of antibiotics and a decrease in morbidity and mortality associated with antibiotics, especially in extremely premature infants during their lengthy hospital stays. However, further studies are needed to demonstrate such adherence in Europe. In the United States, Ho et al. reported a significant gap between the recommendations of the CDC and the policies of NICUs to improve the use of antibiotics [23]. In contrast, our study suggested that in Europe, adherence to clinical guidelines was high for EOS and LOS, but lower for NEC.

As described in the previous literature, the use of neonatal sepsis risk calculators is still limited [15,36,37]. Similarly, there is high variability in the laboratory tests used in the management of neonatal sepsis. However, the combination of blood culture and CRP is the most frequently used test.

Leroux et al. identified considerable variability among French NICUs regarding both the loading and maintenance dose of antibiotics used [38]. They also described significant differences in aminoglycoside administration intervals. In contrast, we did not observe a high degree of variability in the dose and administration intervals of the different antibiotics. Most NICUs follow the pharmacological clinical guidelines of their country or the indications of reference publications, such as Neofax [39]. Nevertheless, it has been challenging to establish a mean dose value since the response was made in free text mode. Furthermore, almost every responder indicated that doses and administration intervals vary according to gestational age, days of life, and severity of infection.

Litz et al. analyzed the management of early- and late-onset neonatal sepsis in German NICUs through a survey that included up to 80 regional and university neonatal units. Their findings showed that the antibiotics used by these units are similar to those used in other European NICUs that participated in our study. They also noted that German NICUs did not perform an LP in all neonates with infection. Instead, they limited its use to cases in which meningitis was suspected (65%). Furthermore, they reported that more than 90% of the participating NICUs monitored the therapeutic levels of gentamicin or vancomycin used for LOS. In contrast, our findings were lower in these parameters. However, it should be noted that our survey involved NICUs from all of Europe and included EOS and LOS [24].

Blood culture remains the standard diagnostic criterion for neonatal sepsis. However, a low sensitivity in neonates, the small volume collection which may lead to a false negative, the risk of contamination during sample collection which may lead to a false positive, and the potential of delayed results push clinicians to seek other methods for early diagnosis [40,41,42,43,44,45,46,47,48,49]. Even if the literature supports the use of rapid methods with high sensitivity and specificity (Cytokines, PCR based methods) or the use of calculators such as Kaiser Permanente EOS, blood culture is still the standard of care for detecting neonatal sepsis in association with CRP or PCT [42]. We speculate that the use of markers with which the neonatologist feels confident limits the use of these new diagnostic tests. In addition, these novel tests are not well-integrated within their out-of-hours laboratories, and the need for clear reference values for decisions on antibiotic administration further hampers their use.

The policies on maternal screening for GBS were found, as expected, to be not homogeneous. About half of the NICUs participating in our survey reported performing microbiological detection of GBS in all pregnant women at the end of pregnancy. Women eligible for intrapartum antibiotic prophylaxis to prevent vertical GBS transmission are generally identified through two strategies: universal culture-based screening for GBS colonization or identifying clinical risk factors for GBS transmission (i.e., prolonged rupture of membranes, bacteriuria, an earlier child with GBS EOS, and maternal fever) [50]. In some European countries, such as the United Kingdom or The Netherlands, universal GBS screening is not recommended in pregnant women. Instead, they only perform GBS screening for situations with risk factors [51,52]. In our study, NICUs from Turkey, The Netherlands, Germany, Sweden, and Denmark claimed that they were not performing universal screening, in contrast to NICUs from Italy, France, Belgium, Portugal, and Spain. Therefore, there is still a lack of consensus in European countries regarding the prevention of perinatal GBS infection [53,54]. A recent meta-analysis showed that universal screening protocols were associated with lower rates of early-onset GBS disease when compared to risk-based protocols [50].

It is still controversial whether an LP should be performed for suspected EOS and LOS with a universal or selective approach [55]. However, our study showed that most European neonatologists decided to perform an LP based on different factors, including the severity of illness, markers of infection, and blood culture results.

We encountered certain limitations in the conduct of this study. However, some of these limitations were derived from the nature of the study. For example, we did not ask about the duration of antibiotic treatment in the survey, a question that may differ from one neonatal unit to another. Furthermore, even though we have insisted that the responses of participating NICUs should come from the director of the NICU or a senior member of staff, we have not been able to control which neonatologists responded to our questionnaire. Thus, the data obtained may not exactly reflect the plan of all neonatologists in that department. In any case, given the high number of responses obtained across different countries, we deem these data as representative of the management of neonatal infection in Europe. This thought process is further strengthened by the fact that participating hospitals included many NICUs with different levels of complexity and numbers of admissions, such as centers with neonatal surgical care. On the contrary, the effect of highly responsive countries such as Italy may have entailed a selection bias, which may have conditioned the results based on the management criteria of scientific societies.

## 4. Materials and Methods

This research project was a cross-sectional study that aimed to identify which antibiotics are used in European NICUs and describe general aspects of neonatal sepsis and perinatal infection management.

The authors developed an online questionnaire on SurveyMonkey (https://www.surveymonkey.com (access available on 1 March 2021, San Mateo, CA, USA), specifically designed to carry out this research. Qualified experts (included in the authorship) reviewed the consecutive versions of content, construct, and face validity until consensus was reached [56].

The respondents were informed that participation in the survey was voluntary and that completing the survey meant approval to participate. This survey consisted of 29 questions in different formats, which respondents could answer in approximately 15 minutes. It was designed in English and distributed to all NICUs. However, there is a wide difference in the organization of care for newborns among different countries in Europe [29]. Moreover, an exhaustive list of European NICUs does not exist [30,31]. To obtain the maximal number of responses, we contacted neonatal units through neonatal scientific networks and societies, social networks, and direct email contact, requesting that they share a web-based survey between the NICUs of their respective countries. In addition, we contacted a different neonatologist in the country when there was no response after two weeks. The survey was disseminated throughout the first trimester of 2021. The head of the NICU or a delegated senior neonatologist was expected to complete the survey. To avoid duplication of information from each neonatal unit, we tried to obtain the response of a single senior neonatologist per NICU (Appendix A). For this, we gave the respondent the option to leave some personal data that would allow us to identify the neonatology unit and hospital on which he or she was offering the answer. With this, we avoided duplication or multiplication of responses from the same unit.

The questionnaire focused on common local practices in the management of neonatal infections, such as the use of antibiotic guidelines and adherence to said guidelines, the use of biological markers, practices in obtaining CSF, the use of oral antibiotics, and the use of antifungal prophylaxis. At the end of the questionnaire, we obtained information about the neonatologists’ knowledge regarding the approved regulations for antibiotics in their country. To compare results from institutions with different resources and levels of complexity, we stratified the data according to NICU level (Appendix A, question 2). Moreover, we stratified NICUs depending on the number of total admissions and the number of admissions with a birth weight below 1500 g in 2020. In some questions, the neonatologist could only choose one option throughout the questionnaire, but they could select one or more answers from others. This information can be consulted in a section of the Appendix A for this article (Appendix A).

Once the survey was closed, it was processed using the generated Excel file. First, duplicate responses from the same hospital were identified, and only one was accepted; this process was evaluated by two researchers (F.G., C.A.). Second, 109 variables were generated to facilitate the analysis of responses. Questions 18, 19, and 20, referring to the combination of antibiotics used in EOS and LOS or NEC, generated an auxiliary variable to define the combination. The same was done for questions on infection markers (questions 21 and 22). Finally, inconsistent responses were identified, particularly for variables corresponding to antibiotic labeling. A flowchart of the cleaning process is shown in Figure 1.

Quantitative variables were analyzed by calculating their central tendency. Next, categorical variables were analyzed by calculating their frequency and percentages. Finally, the statistical relationship between categorical variables was determined using the chi-square test. The statistical package IBM SPSS Statistics version 24 was used for statistical analysis.

## 5. Conclusions

In this European survey, we identified high homogeneity in the empirical treatment of EOS. In contrast, there was substantial variability in LOS and NEC. Adherence to local clinical guidelines was also reported to be high. In addition, the two most prescribed antibiotics were ampicillin and gentamicin. CRP is most commonly used to diagnose neonatal sepsis and decide antibiotic treatment suspension.

Our findings could improve the treatment of neonatal sepsis by highlighting areas in need of further improvements, such as implementing new guidelines focused on establishing antibiotic termination and tuning antibiotic therapy for LOS or NEC.

## Figures and Tables

**Figure 1 antibiotics-10-01046-f001:**
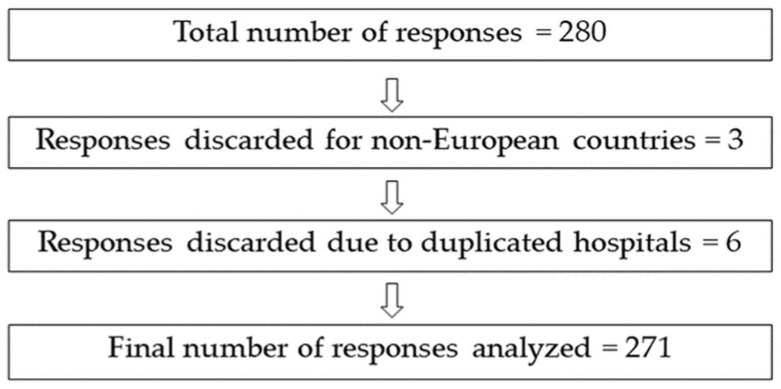
Graphic representation of discarded responses from the survey and total number of analyzable responses.

**Figure 2 antibiotics-10-01046-f002:**
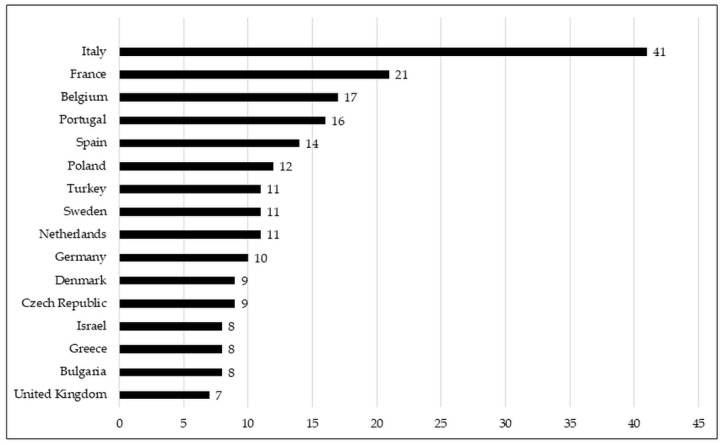
Graphic representation of the number of responses to the survey per country in which more than 6 neonatology units have answered.

**Table 1 antibiotics-10-01046-t001:** Demographics of responding neonatal units.

	n (%)
Level of care of NICUs, n. (%) Level 1 Level 2 Level 3 Level 3 plus * Mixed	
	6 (2.2%)
	44 (16.2%)
	106 (39.1%)
	102 (37.6%)
	13 (4.8%)
Care of premature newborns under 28 weeks until discharge, n. (%)	
Yes No	226 (83.4%) 45 (16.6%)
Neonatal admissions in 2020, median (IQR)	444 (273; 864)
Number of admissions of babies < 1500 g in 2020, n. (%)	
Less than 30 Between 30 and 50 More than 50	76 (28%) 64 (23.6%) 131 (48.3%)
Intensive care incubators in the NICU, median (IQR)	2 (8; 21)

IQR: interquartile range; n: number; NICU: neonatal intensive care unit. * Level 3 plus refers to those neonatal units with surgery, neurosurgery, or heart surgery. See Appendix A for detailed description of levels of care.

**Table 2 antibiotics-10-01046-t002:** Summary of which antibiotics were used by the consulted NICUs in early-onset sepsis, late-onset sepsis, or necrotizing enterocolitis, respectively.

	EOS n (%)	LOS n (%)	NEC n (%)
Penicillin	62 (22.9%)	9 (3.3%)	6 (2.2%)
Ampicillin	204 (75.3%)	38 (14%)	58 (21.4%)
Gentamicin	208 (76.8%)	92 (33.9%)	96 (35.4%)
Tobramycin	6 (2.2%)	2 (0.7%)	4 (1.5%)
Amikacin	45 (16.6%)	86 (31.7%)	60 (22.1%)
Cefotaxime	13 (4.8%)	76 (28%)	44 (16.2%)
Ceftazidime	2 (0.7%)	17 (6.3%)	15 (5.5%)
Meropenem	2 (0.7%)	42 (15.5%)	64 (23.6%)
Vancomycin	1 (0.4%)	142 (52.4%)	107 (39.5%)
Nafcililin	0 (0%)	0 (0%)	0 (0%)
Metronidazole	0 (0%)	1 (0.4%)	140 (51.7%)
Piperacillin-tazobactam	1 (0.4%)	37 (13.7%)	46 (17%)
Teicoplanin	0 (0%)	11 (4.1%)	8 (3%)
Flucloxacillin	0 (0%)	23 (8.5%)	3 (1.1%)
Oxacillin	0 (0%)	14 (5.2%)	0 (0%)

EOS: early-onset sepsis; LOS: late-onset sepsis; n: number; NEC: necrotizing enterocolitis; NICU: neonatal intensive care unit.

**Table 3 antibiotics-10-01046-t003:** Biological markers and laboratory tests used to diagnose neonatal sepsis and decide to suspend antibiotic treatment.

	For Diagnosis of Sepsis n (%)	To Stop Antibiotics n (%)
C reactive protein	264 (97.4%)	238 (87.8%)
Procalcitonin	135 (49.8%)	79 (29.2%)
Cytokines	31 (11.4%)	7 (2.6%)
PCR base methods	46 (17%)	13 (4.8%)
DNA microarray-based methods	12 (4.4%)	6 (2.2%)
Blood culture	258 (95.2%)	209 (77.1%)
HeRo monitoring	20 (7.4%)	9 (3.3%)
Full blood count	32 (11.8%)	17 (6.3%)

**Table 4 antibiotics-10-01046-t004:** Questions about the decision to perform a lumbar puncture and cerebrospinal fluid examination in a clinically septic neonate, n (%). Participants could answer one or more options.

Questions	Responses	
	n	%
I always perform a lumbar puncture if the newborn is clinically stable	43	15.9
I decide to perform a lumbar puncture based on the patients’ history, risk factors, clinical features	191	70.5
I decide to perform a lumbar puncture depending on blood markers (CRP, for example)	62	22.9
I decide to perform a lumbar puncture depending on the positivity or negativity of the blood culture	94	34.7

## Data Availability

All data relevant to the study are included in the article or uploaded as Appendix A. Additional data are available upon request.

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
