# Peer review of "Variations in Antibiotic Use and Sepsis Management in Neonatal Intensive Care Units: A European Survey"

_antibiotics, 2021, doi:10.3390/antibiotics10091046_

Round 1

Reviewer 1 Report

This is an interesting study with relevant findings. My comments are below:

Abstract:

  • State clearly the aim of this study
  • For this statement “Most of the units had a written clinical guideline for the management of sepsis”, insert the percentage for these units.
  • For this statement “management of neonatal sepsis in European NICUs shows a wide diversity of clinical guidelines with high self-reported adherence to them”- where is the self-reported adherence info? Currently the conclusion do not tie well with presented findings.

Introduction:

  • Line 52, is it 30 million cases? If yes, remove the space between 3 and 0.
  • Lines 83-87 require further clarification- briefly explain/elaborate on the existence of clinical guidelines in this statement “significant variability in the dosing of commonly used antibiotics in the neonatal period [26,27] and the existence of clinical guidelines”.

Results:

  • Line 106: Most responding units were level 3 (208; 78.2%)- please define described levels where relevant?
  • Lines 124-126: “Regarding GBS detection in mothers at the end of pregnancy, 60.5% of the centers performed universal screening, 21% screened selected mothers, and 18.4% did not test for GBS.” Please provide context for the screening policy in the methods section or discussion section- so what is universal screening, screened selected mothers, etc?
  • Line 150: “One-third (29.9%) of the NICUs never perform therapeutic drug monitoring 150 (TDM) of the antibiotics administered in their unit.” Any reasons/ context for not performing TDM?

  • Line 175: delete the following statement: “As we mention in the methods, respondents could choose one or more options in this question”.

Discussion:

  • Line 182: This statement “Detailed rational knowledge of antibiotic prescribing patterns should be implemented in clinical practice” does not read well. Either amend it or delete it and start the discussion with next statement, i.e. Judicious use of antibiotics is a key way to….

  • Lines 184-187: so how your findings fit within the WHO recommendations? Please state first you relevant findings and then discuss it in line with the provided recommendation.

  • Line 187: how is this statement relevant to this article “This recommendation can be applied to developing countries [31].”? Please elaborate

  • Line 218: what “these percentages” are you referring to?

Materials and methods

  • Line 315: please state first the name of the platform that was used to administer this questionnaire, and then the URL (website) for it.

  • Provide some information about the development of this survey, e.g. face/content validity, pilot-testing, which language was used/ translated to if any.
  • Please provide a statement on sample size calculations- so what is the total number of targeted population and what percentage your sample represent? Please discuss relevant implications if any.

Conclusion:

  • Lines: 365-367: This statement better to be written into 2 sentences: “The two most prescribed antibiotics were ampicillin and gentamicin, and C reactive protein was most commonly used to diagnose neonatal sepsis and decide to suspend antibiotic treatment.”

Others:

Line 385: authors state the following: “Institutional Review Board Statement: Not applicable”; does this mean that this study did not go through relevant ethics committee? If this study did not go through relevant ethics/approvals, please explain why?

Author Response

Dear Reviewer 1,

Thanks for your comments and suggestions.

Please find attached our responses.

An english certificate from Editage has been obtained.

Regards

Authors

Reviewer 2 Report

The aim of the article submitted by Garrido et al., was to evaluate the use of antibiotics in the treatment of neonatal infections in European NICUs, as well as to characterize the management of sepsis and perinatal infections across Europe.

The article is interesting, well written and clear. Only a few minor points should be revised:

  • Table 1, table 2, , table 4 should be formatted
  • Line 164 “as described in the literature”, the word “the” should be removed
  • Line 169 “establish” should be changed to “established”. Please check typos throughout the text
  • Lines 256-258: the sentence should be revised
  • Lines 260-261 and line 264: “The questionnaire consisted of 31 questions” is repeated twice. However, what was the correct number of questions (31 or 29?)? In Suppl 5 there are 29 questions.
  • The link of the online questionnaire is not available, as the survey is  now closed. What is the reason for reporting the web link in the text? Please explain
  • The authors should check references, and write the Journal names in  abbreviated form. Please, see information in "Instructions for authors"

Author Response

Dear Reviewer 2,

Thanks for your comments and suggestions.

Please find attached our responses.

An english certificate from Editage has been obtained.

Regards

Authors

Round 2

Reviewer 1 Report

Thanks for addressing my comments